# A Short Review of Cavity-Enhanced Raman Spectroscopy for Gas Analysis

**DOI:** 10.3390/s21051698

**Published:** 2021-03-02

**Authors:** Christian Niklas, Hainer Wackerbarth, Georgios Ctistis

**Affiliations:** Institut für Nanophotonik Göttingen e.V., Hans-Adolf-Krebs-Weg 1, 37077 Göttingen, Germany; hainer.wackerbarth@ifnano.de

**Keywords:** Raman spectroscopy, gas analysis, cavity enhancement

## Abstract

The market of gas sensors is mainly governed by electrochemical, semiconductor, and non-dispersive infrared absorption (NDIR)-based optical sensors. Despite offering a wide range of detectable gases, unknown gas mixtures can be challenging to these sensor types, as appropriate combinations of sensors need to be chosen beforehand, also reducing cross-talk between them. As an optical alternative, Raman spectroscopy can be used, as, in principle, no prior knowledge is needed, covering nearly all gas compounds. Yet, it has the disadvantage of a low quantum yield through a low scattering cross section for gases. There have been various efforts to circumvent this issue by enhancing the Raman yield through different methods. For gases, in particular, cavity-enhanced Raman spectroscopy shows promising results. Here, cavities can be used to enhance the laser beam power, allowing higher laser beam-analyte interaction lengths, while also providing the opportunity to utilize lower cost equipment. In this work, we review cavity-enhanced Raman spectroscopy, particularly the general research interest into this topic, common setups, and already achieved resolutions.

## 1. Introduction

Gas analysis in both industrial and civilian environments has always been important, yet gained more momentum in view of stronger environmental safety regulations due to climate change in recent years. Commonly used gas sensors work electrochemically or are semiconductor-based. They are very sensitive on a specific analyte with a disadvantage of strong cross-sensitivities to other analytes. Multi-gas analysis is utilized with so-called electronic noses, which are used to detect odorous gases [1,2,3]. These noses are sensor arrays of electrochemical and semiconductor-based sensors, which are chosen according to the odours to be detected and its corresponding volatile organic compounds [4]. Though multi-gas analysis is possible with these sensor arrays, these face challenges since the necessity of prior knowledge of the composition for the sensor to work properly is essential. Optical gas sensing, on the other hand, is a very powerful tool handling the aforementioned difficulties. Furthermore, it is mostly non-invasive and, thus, can be used even in harsh and hazardous environments, as well as even from long distances. The most common approaches are infrared-based absorption technologies, either without spectrometer, such as non-dispersive infrared absorption (NDIR) [5,6,7,8,9], or with full spectroscopic analysis, including (Fourier-transform) infrared absorption spectroscopy (FTIR) [10,11,12], tuneable diode laser absorption spectroscopy (TDLAS) [13,14,15,16,17], photoacoustic spectroscopy [18,19,20,21], or cavity ring-down spectroscopy [22,23,24,25,26,27]. They all provide very high resolutions and limit of detections (LOD), down to ppb level, as they face very large absorption cross sections due to the excitation of fundamental vibrations in the infrared spectral region. In the wake of their development, data analysis techniques (chemometry) also evolved [28,29,30,31,32,33,34,35], extending the capabilities to also include multiple specimens. Yet, besides these strengths, infrared absorption techniques have their limitations: First, and most important of all, the detection of homo-nuclear gases, such as N${}_{2}$, O${}_{2}$, and H${}_{2}$, is nearly impossible using infrared absorption techniques. Yet, the detection of these analytes becomes more and more important in different fields, such as biology [36,37] and medicine, e.g., breath analysis and disease detection [38,39,40,41], as well as in energy [42,43,44,45] and mobility. The latter known due to the shift towards hydrogen and electric mobility and the need for in-situ monitoring of the underlying processes for optimization. Second, water absorption has an overwhelming effect and, as it has a large footprint in a broad window in the infrared, can mask the absorption due to other analytes.

An alternative optical method overcoming these deficiencies is Raman spectroscopy. As all gases, with exception of mono-atomic gases, have Raman active oscillations, as long as their polarizability changes due to the oscillation, Raman spectroscopy is the right method for even multi-analyte mixtures in complex environments. Most importantly, the samples do not need further treatment, since water is no hindrance for Raman spectroscopy. Yet, besides the positive aspects, one inherent disadvantage of Raman spectroscopy, limiting its use, is an extremely low scattering cross section. Therefore, detecting trace gases with a very low limit of detection (LOD) is difficult. Improving the scattering intensity is thus a key-issue for gas analysis with Raman spectroscopy. There have been, therefore, various efforts to circumvent the low scattering cross sections by increasing the light-matter interaction. Aside from using high-power lasers, field-enhancement methods, such as surface-enhanced Raman scattering (SERS) [46,47,48,49,50,51], the selection of an excitation wavelength in resonance with an electronic transition of the molecule in resonance Raman (RR), or the exploitation of non-linear effects, as it is done in coherent anti-Stokes Raman scattering (CARS) [52,53,54], are the most used techniques.

Another method to improve the scattering intensity, thereby enabling the use of low-cost, low-power lasers are cavity-enhanced techniques, i.e., cavity-enhanced Raman spectroscopy (CERS), where a cavity or multi-pass cell is used to enhance the interaction length and the intensity of the laser beam. Due to an increasing amount of publications in the last few years and the increasing number of different and complex setups, as well as an increasing amount of applications, we review here the current state of cavity-enhanced Raman spectroscopy and its applications, especially in the context of gas analysis. Different setups, with differing cavity design and working principle, are evaluated regarding their gain. Furthermore, current limits of detection achievable with CERS are compared and evaluated.

This review is organized as follows. First, the basic principles of the Raman scattering and theory are introduced, followed by the principles and techniques of enhancing the Raman scattering. Next, we present the basic principles of, as well as necessities for, gas analysis using cavity-enhanced Raman spectroscopy. The main part of the review compares the performance of different CERS setups with each other and with state-of-the-art sensors regarding capabilities, advantages, and disadvantages in gas sensing. Finally, the possibilities for implementation in Internet of Things (IoT) applications are discussed.

## 2. Raman Theory

Raman scattering and consequently spectroscopy was first experimentally observed by Raman in 1928 [55], after being theoretically predicted by Smekal in 1923 [56]. Classically, the Raman effect can be described as an inelastic scattering process of photons and molecules. The electric field of the photon interacts with the hull electrons of the molecules and induces a dipole moment, which oscillates with the light frequency. Besides the classical approach, Raman scattering can be described by quantum theory. Here, the occurring scattering processes for photons and molecules and their transitions, as well as infrared absorption, are shown in Figure 1.

The main scattering process of light with molecules is called Rayleigh scattering. Here, the molecule is excited to a higher energy virtual state and relaxes to the ground state, so no change of energy occurs. The Raman Stokes scattering is a process, where the molecule is excited to a higher energy virtual state, but does not relax to the ground state but to a vibrational state, the molecule vibrates now, resulting in emitting a lower energy photon compared to the absorbed one. The difference between the incident photon and the outgoing photon is called the Raman shift. This shift is independent of the excitation wavelength. Comparing the intensities of the Stokes and Rayleigh scattering, Stokes is by orders of magnitude weaker since the scattering cross section of Stokes is much smaller than for Rayleigh. The last scattering process shown in Figure 1 is anti-Stokes scattering. Here, a molecule, which is already in an excited state, is excited towards a virtual state of higher energy and relaxes afterwards to the ground state. Therefore, a blue shift occurs, as the emitted photon has a higher energy than the incident photon. The occurrence ratio between Stokes scattering and anti-Stokes scattering is heavily dependent on the conditions of the used setup and the analyte. As anti-Stokes scattering needs already excited molecules, Stokes scattering has a higher probability at ambient conditions, since the occupation probability of a state is governed by Boltzmann distribution, and, thus, nearly all molecules are in ground state. Therefore, for gas sensing applications at ambient conditions, solely the Stokes scattering intensity is taken into account.

Due to the accessibility of more powerful lasers, Raman spectroscopy has become more attractive, as the Raman signal is directly proportional to the incident intensity, which, on the other hand, leads to higher resolution. The intensity of the Raman signal $I_{Raman}$ is dependent on various variables and constants [57]:(1)$$I_{Raman}=\varepsilon \;I_{Laser}\;\frac{N}{V}l_{eff}\;\sigma_{Raman}.$$

Here, ε is the total detection efficiency, $I_{Laser}$ the incident laser intensity, N/V the amount of molecules inside the illuminated volume, and $l_{eff}$ the effective interaction length. The last variable $\sigma_{Raman}$ is the analyte-specific Raman cross-section, which behaves as [57,58]
(2)$$\sigma_{Raman}\propto {\left(\frac{1}{\lambda_{0}}\right)}^{4},$$
where $\lambda_{0}$ is the excitation wavelength. It is evident that lower wavelengths have far greater Raman yield. Nonetheless, lower wavelengths also face some challenges. As fluorescence has the higher generation efficiency than the Raman scattering signal, wide fluorescence signals can conceal Raman peaks, especially roto-vibrational Raman peaks. Furthermore, due to technological, physical, and financial limitations, wavelengths below visible are the exception rather than the norm, although, in recent years, UV-lasers are becoming more common. Laser systems, such as free electron lasers, are not in any way efficient for commercial gas sensing due to their scarcity. The variables ε and N/V are not variable for Raman signal enhancement as the total detection efficiency is limited by the used detector and applied detection geometry, and N/V describes the specimen for detection. One can try to change either the intensity of the laser, the effective interaction length, or the wavelength and, thereby, the Raman cross section. Choosing lower wavelengths leads to higher fluorescence and small wavelength high power lasers have higher financial costs, rendering an industrial application improbable. The effective interaction length $l_{eff}$ can be enhanced by various efforts, by measures, such as using multi-pass cavities, or longer sample containment. The intensity of the laser can be increased by either using high power lasers, i.e., increasing incident power, or by power enhancement methods, as achieved by using external cavities.

As high power lasers are financially expensive, low-cost lasers are more suited for industrial applications. Though not evident in these equations, but important nonetheless, is the bandwidth of the laser. Here, a narrow bandwidth is better suited for Raman spectroscopy, since a larger bandwidth also broadens the the Raman signal masking thereby excitations lying close together. Furthermore, it is possible to resolve rotational spectra with narrow bandwidth excitation. In recent years, several enhancement methods have been developed and are still being improved to enhance the Raman signal intensity. They will be introduced in the following section.

## 3. Raman Enhancement Methods

Up to now, several Raman enhancement methods have emerged, ranging from technical easy implementation into existing Raman setups, such as Resonance Raman spectroscopy [59,60,61], or rather complex ones, such as coherent anti-Stokes Raman scattering (CARS) spectroscopy [52,53,54,62]. Resonance Raman spectroscopy is based on the respective resonances of the analyte, as it uses the same excitation wavelength as the electronic transition of the specific analyte. This method’s setup only differs from an usual Raman setup in the excitation laser. For resonance Raman, a wavelength tuneable laser is needed and depending whether a single or a compound of analytes is used, a multitude of lasers is required to satisfy various electronic transitions. Therefore, gas compounds with low traces of gases require a multitude of lasers to enable Resonance Raman spectroscopy, which is too expensive to be used in non-laboratory environments.

CARS spectroscopy was first mentioned in 1974 by Begley et al. [52]. Instead of Stokes scattering, anti-Stokes scattering is utilized. To efficiently generate anti-Stokes, four-wave mixing is utilized, as the difference of two pump beams is chosen to match the Raman frequency, therefore generating anti-Stokes scattering. CARS needs two pulsed laser systems, instead of one laser used by other Raman techniques, making CARS setups rather complicated and monetarily expensive.

Another method to increase the Raman signal is surface-enhanced Raman scattering (SERS). Here, metal nanostructures enhance the electrical field at their surface or edges. The analyte needs to be close to areas of high electromagnetic fields. The challenge of gas analysis is to bring the molecules from the bulk to these specific areas. Even though the SERS-method has shown its capability for single molecule detection [63], it cannot be used for bulk analysis, as the molecules must be very close to the surface, rendering it difficult to implement in industrial applications.

Another potent possibility for the enhancement of the Raman signal of gases is fiber-enhanced Raman spectroscopy (FERS). Here, both the laser intensity and the interaction length are enhanced for a larger Raman signal, similar to CERS. It is based on hollow-core fibers and gained additional traction because of hollow-core photonic crystal fibers. As photonic crystal fibers do not rely on total internal reflection but on the photonic bandgap, it is possible to use the hollow core as the analyte chamber [64]. Here, light and the analyte are coupled into the fiber, and, due to multiple reflections inside the fiber, the interaction length is enhanced compared to the free space trajectory, while also having the ability to stay compact as the fiber can be wound onto a drum or core, thus allowing multiple meters of interaction length. Furthermore, as those hollow cores in the case of photonic crystal hollow core fibers are only micrometers in diameter, the laser power is focused inside the hollow core, therefore resulting in a electrical field magnification, and consequently resulting in a higher Raman signal. With a low power laser diode and a hollow-core photonic crystal fiber, it is possible to detect atmospheric gases, such as O${}_{2}$ and N${}_{2}$ under ambient conditions, and a CO${}_{2}$ resolution of 1.6% was achieved [65]. Hanf et al. have demonstrated with their hollow-core photonic crystal fiber setup limits of detection in the ppm range [66]. Here, Hanf et al. were able to achieve an LOD for CH${}_{4}$ as low as 0.2 ppm. One big challenge is a large background of the Raman spectrum due to the fiber itself but can be overcome by the use of the right pinhole conjugated to the fiber output tip [67]. Additionally to hollow-core fibers, metal-coated capillaries can be used as a compact multi-pass cell. Here, the inside of the capillary is coated with a metal, which depends on the desired excitation wavelength [68]. A similar method to increase Raman scattering is cavity-enhanced Raman spectroscopy (CERS), which will be discussed in the next section. The mentioned Raman enhancement methods are shown in Table 1.

## 4. Cavity-Enhanced Raman Spectroscopy

One of the first published uses of CERS was done in 1967 by Weber et al. [69]. Here, a so-called Raman tube equivalent to a multi-pass cell was put inside a HeNe laser to enhance the Raman signal. Ten years later, in 1977, Hill et al. [70] used a multi-reflecting cell to enhance the Raman signal. The name itself, cavity-enhanced Raman spectroscopy, is first mentioned in an article in 1995 [71]. Here, water droplets were used as microcavities to enhance the Raman signal to give the information of the identity and amount of species present in the droplets themselves. This has led to the so-called cavity-enhanced droplet spectroscopy [72].

In the first years, between 1995 and 2000, only 11 publications can be found regarding CERS, which are all about cavity-enhanced droplet spectroscopy. Up until 2010, an additional 37 publications regarding CERS were published. Only in recent years has CERS gained traction, with over 200 publications in the span of the last decade. Nonetheless, cavity-enhanced droplet spectroscopy has the majority of publications regarding CERS. Furthermore, FERS, as a subcategory of CERS, has gained momentum due to advances in fiber technology, especially in the field of hollow-core photonic crystal fibers [67,73].

The basic concept of cavity-enhanced Raman spectroscopy is comparable with other cavity enhancement methods, such as cavity-enhanced absorption spectroscopy. Here, the interaction pathlength is increased to enable the detection of small analyte concentrations. This can be done using so-called multi-pass geometries, where, through reflections, light passes the same test volume multiple times. Cavity-enhanced Raman spectroscopy can utilize both a longer interaction length and the optical resonator effect, where an external cavity acts as a power build-up cavity (PBC). Here, depending on the quality of the cavity mirrors, the cavity enables the use of a cost-efficient low-power laser for Raman measurements.

### 4.1. Basic Cavity Concepts

Here, the basic working concepts of cavities will be outlined for a better understanding of the mechanisms leading to enhancement in CERS. A cavity is generally characterized by its two mirrors and the material between those mirrors. The theoretical gain *G* of a cavity in resonance is given by [74]
(3)$$G_{cavity}=\frac{4T_{1}}{{\left(T_{1}+T_{2}+A\right)}^{2}}.$$

Here, $T_{1}$ and $T_{2}$ are the respective transmittivities of the cavity mirrors, used for in- and out-coupling, and *A* is the round-trip loss due to absorption. For CERS measurements, *A* is dependent on the analyte; therefore, only the mirrrors can be effectively used for the enhancement of the incident laser power. Another variable to describe an optical cavity is the finesse *F*. It is described by the free spectral range, which is defined as the spectral difference between two spectral maxima Δλ, and the full width at half maximum (FWHM) of a single maximum spectral line δλ:(4)$$F=\frac{\Delta \lambda }{\delta \lambda }.$$

For high reflectivities of the cavity mirrors and a small damping, the finesse can be described through the reflectivity *R*:(5)$$F=\frac{\pi \sqrt{R}}{1-R}.$$

If the reflectivity for both mirrors differs, *R* is the effective reflectivity. The finesse itself is a quality indicator for an optical resonator: the higher the finesse, the higher the buildup power inside the resonator. An alternative way to describe a resonator is the so-called quality factor *Q* an indicative of the energy loss inside the cavity. It can be calculated as [74]:(6)$$Q=\nu_{0}T_{roundtrip}\frac{2\pi }{I_{loss}}.$$

Here, $T_{roundtrip}$ is the time it takes for the laser light to take one round trip, and $I_{loss}$ the intensity loss during that round trip. Both *F* and *Q* are indicators for the possible intracavity power. The enhancement factor β is calculated with the reflectivity *R*:(7)$$\beta =\frac{1}{1-R}.$$

In the case of resonant CERS, there are some geometrical key numbers. The cavity itself has a free spectral range depending on the refractive index *n* inside the cavity and the distance *l* between the cavity mirrors:(8)$$\nu_{FSR}=\frac{c_{0}}{2\cdot l\cdot n}.$$

The linewidth Δν, which limits the usage of the laser and the effective in-coupling can be estimated by the finesse from Equation (Equation 5) and the free spectral range:(9)$$\Delta \nu =\frac{\nu_{FSR}}{F}.$$

Furthermore, depending on the used cavity as seen in Figure 2, the best possible analyte volume differs. In cavities with a focal point, and the analyte volume in the resonator with the highest laser power, which is the focal point, the beam radius $W_{cav}$ is one key value: (10)$$W_{cav}=\sqrt{\frac{\lambda l}{2\pi }\sqrt{2\frac{\left|R\right|}{l}}-1}.$$

Here, *l* is the distance between the cavity mirrors, and *R* is the curvature of the mirrors. This equation is for the special case of a symmetric spherical mirror. Furthermore, the cavity is additionally described by the Rayleigh range:(11)$$z_{0,cav}=\frac{l}{2}\sqrt{2\frac{\left|R\right|}{l}-1}.$$

This describes the distance along the optical axis, in which the beam radius grows factor $\sqrt{2}$. Furthermore, the theoretical gain can be roughly calculated by the reflectivity of the used mirrors:(12)$$g\approx \frac{1}{1-R}.$$

Both values $W_{cav}$, as well as $z_{0,cav}$, fully describe the analyte volume of the cavity, where the main laser power resides, which can be estimated by the gain *g*, therefore being essential to CERS.

### 4.2. Cavities and Locking Methods

CERS spectroscopy has seen a rise in applications and research in recent years. Various setups and measurement methods, as well as means to keep a cavity stable, have thereby been utilized, which will be showcased in the following.

#### 4.2.1. CERS Setups

Cavity-enhanced Raman spectroscopy setups can be sub-divided into stable and non-stable resonators, which themselves can be further sub-divided depending on the number of mirrors used and other variants. Most common setups built use non-resonant multi-pass cavities and Fabry-Perot cavities, which are already utilized in cavity-enhanced absorption spectroscopy. Furthermore, integrated cavities have been proposed [75,76,77]. These type of cavities are robust due to their construction and can be used, e.g., for interplanetary measurements [78]. These cavities will be explained further in a later chapter. Cavities themselves are known from laser technology and can have various shapes, which offer different advantages and disadvantages. Here, the mirrors can be in a plane parallel, concentric, confocal, hemispherical, or concave-convex setup, as long as the setup satisfies the stability condition $0\leq g_{1}g_{2}\leq 1$ with
(13)$$g_{1}g_{2}=\left(1+\frac{L}{r_{c1}}\right)\left(1+\frac{L}{r_{c2}}\right),$$
where *L* is the distance between both cavity mirrors, and $r_{cx}$ is the curvature of the respective cavity mirror [79]. This stability condition is shown in Figure 2.

Additionally, some of the most used resonators, concentric, confocal, and plane-parallel are shown there, as well. A resonator configuration should be stable, even with small length changes, e.g., created by mechanical vibrations. The confocal cavity meets these criteria and is, therefore, a favorable setup, as small changes of the length do not force the cavity outside the stable region. Another important factor for the consideration of the right cavity is its linewidth Δν, which describes the possible linewidth of the resonant light inside the resonator [74]:(14)$$\Delta \nu =\frac{\nu_{FSR}}{F}.$$

Here, $\nu_{FSR}$ is the free spectral range (FSR) of the cavity’s TEM${}_{00}$ mode, and *F* is the finesse of the cavity. The free spectral range can be calculated as:(15)$$\nu_{FSR}=\frac{c}{2L},$$
where *L* is the length of the resonator. Besides these commonly used cavities, there is a multitude of other geometries, which can be used, as long as they fulfil the stability criteria if a resonant cavity is desired. For example, Wang et al. used a v-arm resonator with 3 mirrors for CERS [80]. In addition to the resonant resonators, multi-pass cells can be used. Here, focal points can be used to enhance the laser intensity; otherwise, only the interaction length is inside the cavity is enhancing the signal. Furthermore, the signal detection is important. Examples of commonly used Fabry-Perot cavities and multi-pass cavities are shown in Figure 3. The mirrors are concave, but plane mirrors can also be used, depending on the desired cavity geometry.

Both Figure 3a,b are Fabry-Perot cavities, but the setups differ in their Raman signal collection path. The cavity setup in Figure 3a can be used either in forward or backward scattering. A notch or dichroic filter is mandatory, as, otherwise, the excitation laser is too strong, i.e., the Rayleigh scattered light, for the Raman signal to be detected, especially for small Raman shifts. The setup in Figure 3b shows a $90^{\circ }$ detection angle. Here, a third optional mirror is shown, which can further enhance the Raman signal by collecting the back scattered light. In the best case, the Raman signal can thereby be doubled. The last setup depicted in Figure 3c, shows a multi-pass cavity. This setup is non-resonant, so mainly the enhancement of the interaction path length yields an enhancement of the Raman intensity. Using spherical mirrors, focal spots are created where an enhancement of the laser power is available and, thus, further enhancement of the Raman signal. In this setup, a third optional mirror can be used, as well, as shown in Figure 3b. Under the assumption, that the scattering is omnidirectional due to gas movement and from a geometrical point of view, the $90^{\circ }$ setups with an additional collection mirror yield a higher Raman gain. Whereas the forward or backward scattering is mainly dependent on the size of the lens after the second cavity mirror, the efficiency of the $90^{\circ }$ geometry can be doubled by utilizing an additional mirror. Although, in reality, even in analytes, such as gases, there is a preferred scattering direction due to the polarized incident light, there is not a great difference in this regard whether one uses forward or $90^{\circ }$ detection. As mechanical vibrations can offset the cavity length, different precautions need to be taken into account to ensure stable enhancement for those cavities relying on resonant setups. These precautions will be discussed in the following subsection.

#### 4.2.2. Locking-Methods

In recent years, different methods have been utilized to control resonant behavior of an external cavity. These methods shall be only briefly discussed here. A more detailed treat of the subject can be found in the review of Wang et al. [81].

The most commonly used method is the Pound-Drever-Hall (PDH) technique, which has ever since become a standard for laser locking [82,83,84]. For a better understanding of this technique, a look at the frequency-based transmission of a cavity is taken in Figure 4. Furthermore, the reflected intensity of a cavity is shown in Figure 4.

Here, three resonances of a lossy resonator are shown exemplary. Either the frequency of the laser itself can be modulated to match the frequency of the cavity or the width of the cavity can be changed to the laser frequency, thus changing the cavity’s eigenfrequency. Normally, the former method is used, which results in a changed intensity. This can be used as a feedback to hold the frequency constant. Yet, as the intensity can fluctuate, this method is not optimal as a frequency lock. As an alternative the reflected intensity in Figure 4 is taken into account. Here, the frequency of laser and cavity are locked when the reflected light is zero. But, rather than looking at the raw reflected intensity, the derivative of the reflected intensity is considered. In this configuration, a frequency change results in a signal showing, whether one is above or below resonance. So, in conclusion, PDH utilizes the change of reflected intensity versus the change of frequency to lock the laser frequency in resonance to the cavity frequency. A schematic for such a setup can be seen in Figure 5. A more detailed description about PDH has been given by Black [85].

Another frequently used mode-locking method is optical feedback frequency-locking (OFFL). Here, analogous to optical injection locking, semiconductor lasers can be used. Contrary to the PDH technique, a processed signal is not used as a feedback for the laser source, but rather the intracavity frequency itself. Step by step, this locking method can be described as follows: The laser, which can have a broader output spectrum, is coupled into an optical cavity. This cavity works as both a frequency standard, as well as a master laser, giving rise to a specific frequency, which, then, as the output of the cavity, is coupled into the diode laser source. One, thus, creates, through optical injection locking, a slave laser, in which laser output depends on the cavity. In this method, there must not be a direct reflection from the cavity to the laser source, as this would disable optical injection locking. This can be guaranteed by using optical isolators, which are already commonly used for the prevention of back-scattering into lasers. A schematic for this locking method can be seen in Figure 6.

This locking method compared to the PDH is cheaper, as no costly electronics equipment is needed. Otherwise, both PDH locking, as well as OFFL, are suitable for controlling the cavity stability. An overview of the mentioned locking methods and their advantages and disadvantages are shown in Table 2.

### 4.3. Integrating Cavities and Microcavities

Besides the aforementioned classes of cavities, there are other special cases, such as integrating cavities and microcavities, which will be discussed briefly. Integrating cavities are no new concept, as in 1970 Elterman described Integrating Cavity Spectroscopy, where an integrating cavity is used to determine the absorption of an unspecific sample [86]. Here, the cavity leads to the independence of scattering within the sample, reflectivity of sample surface, and the geometry of the sample. One problem for integrating cavities used to be a low reflectivity as a significant signal enhancement needs reflectivities higher than 99.5%. In the last few years, manufacturing capabilities increased enormously, so one is able to create high-precision monolithic microcavities. This possibility leads to simplified setups compared to ordinary CERS setups, as the core feature of CERS, the cavity, is more robust and can be utilized for commercial applications. Thereby, applications even for space research have been proposed [78]. Furthermore, integrating cavities can be scaled down in such a way, that microcavities are formed, where quantum effects take the lead role in the amplification of radiation. One of the most important effects is thereby the Purcell effect, describing the probability for the enhancement of spontaneous emission of an emitter inside a cavity [87,88,89]. The higher emission rate can thereby be described by the Purcell factor:(16)$$F_{p}=\frac{3}{4\pi^{2}}{\left(\frac{\lambda_{0}}{n}\right)}^{3}\left(\frac{Q}{V}\right).$$

Here, $\frac{\lambda_{0}}{n}$ is the wavelength within the cavity material with a refractive index of n, *Q* is the quality factor of the cavity, and V its mode volume. It is evident that the ratio of Q/V has to be high to gain a large Purcell factor. It is also the point where one can adjust the system. A small mode volume, corresponding to high spatial confinement, and, additionally, a high quality factor, corresponding to temporal confinement, translates to a large enhancement. Therefore, high reflectivities of the mirrors and small bandwidths of the laser are desirable.

## 5. Evaluation of Recent CERS Setups

As previously mentioned, cavity-enhanced spectroscopy is an already established method and is utilized in various areas. To give an overview of cavity-enhanced Raman spectroscopy and its development, a study of this topic was conducted with Google Scholar. For this study, the exact search term “cavity-enhanced Raman spectroscopy” was used. This search yielded 272 results from 1995 to 2019. Out of these 272 results, 46 are directly tied to gas spectroscopy, whereas the other publications are related to Droplet spectroscopy or others. In Figure 7, these results are shown separated for each year, between “cavity-enhanced Raman spectroscopy”, in general, and gas spectroscopy, in particular.

As already mentioned in the introduction to this review, the earliest publications regarding CERS are mostly related to droplet spectroscopy. Only in recent years has gas spectroscopy been directly used in conjunction with CERS. These publications are going to be analyzed further regarding their setup and analytes.

### 5.1. Enhancement for Different Setups

In this section, recent publications regarding CERS are discussed and compared. Here, papers are assigned regarding their cavity type and geometry. For each cavity type, enhancement factors are compared, and their difference is discussed. In Table 3, the gain for different non-resonant multi-pass cells is summarized.

As can be seen, different geometries were used, as both concentric, as well as confocal mirror setups, were utilized. The highest Raman gains were achieved by using an additional mirror, as it doubles the collection efficiency in those setups than without it. Furthermore, Li et al. [90] report the highest Raman gain using a multi-pass setup. Li et al. used a 532 nm laser with a power of 200 mW, achieving powers, at both focal points of their near confocal setup, exceeding 9 W with 50 multiple passes.

In Table 4, resonant CERS setups are listed. Here, the locking technique, the direction of the Raman detection, and the resulting Raman gain are shown.

The highest Raman gain visible here belongs to the Group of Zaitsu et al. [96], reporting a Raman gain of 6000. Zaitsu et al. observed under optimized conditions an enhancement of a CARS signal. As previously mentioned, CARS setups tend to be more complicated and expensive and, therefore, are not suitable for this study. Instead, the second highest gain will be discussed. Here, a gain of 5900 was achieved by Friss et al. [98]. This was achieved by using a laser at 1064 nm with an incoupling power of 3.7 mW enhanced to 22 W intracavity power. These values are compared with Sandfort et al., as both used the same cavity-locking technique, PDH, as well as the same Raman detection geometry [100]. Sandfort et al. achieved 2.46 W intracavity power at 2.9 mW incoupling power of a 780.2 nm laser. Both could have further enhanced their Raman gain, by applying a third mirror for collection, as shown in Figure 3b. Friss et al. used for their cavity a flat and a concave mirror, translating to a near hemispherical cavity setup with a mirror reflectivity of R=0.99985, whereas Sandfort et al. used two identical plano-concave mirrors with a reflectiviy of R=0.998825 in a near-confocal setup. The difference in reflectivity of only ΔR=0.001 results in a difference of 5049 in Raman gain. Furthermore, the work of Friss et al. can be compared to other work utilizing a different locking or detection setup. Here, Salter et al. used an optical feedback loop and forward detection, utilizing two concave mirrors with a reflectivity of R=0.99988 in a near confocal setup. The setup of Wang et al. [80] is hard to compare to the other setups, as it uses an entirely different geometry in the form of a “V” using three mirrors with a reflectivity of R=0.9992. Nonetheless, this setup still enhances the Raman signal three orders of magnitude. In a previous setup, Wang et al. used CERS with a Raman microscope with a Raman gain of 11.8 [99]. Here, a plane-parallel setup has been used, and Raman light has been collected in a forward geometry. Thorstensen et al. achieved a 50-fold Raman gain with a low cost setup [97]. In conclusion, the different locking techniques or Raman detection geometries have no significant advantages and can be chosen according to the required task.

### 5.2. Benchmark for CERS Setups

As seen in previous sections, CERS as a gas measurement tool has seen a rise in publications. Here, we take a look at the achieved benchmarks, such as limit of detection (LOD), with this methodology. Of course, we have to keep in mind, that various components influence these benchmarks, such as different detectors and different lasers, for example. In Table 5, the LOD of the most common analytes for Raman gas measurement are shown, which are the three atmospheric gases CO${}_{2}$, O${}_{2}$, and N${}_{2}$, as well as the trace gas H${}_{2}$.

Here, Li et al. [90] reached a very low LOD of 16 ppm for CO${}_{2}$, as well as 12 ppm for O${}_{2}$. Wang et al. [80] achieved a comparable LOD with 17.4 ppm for CO${}_{2}$ and 50.7 ppm for O${}_{2}$, although Li et al. used a near confocal multi-pass cell, and Wang et al. used a v-shaped resonant cavity. Both setups used rather long integration times of 1000 s and 200 s, respectively. Furthermore, Li et al. used a 532 nm, 200 mW laser, whereas Wang et al. used a 642 nm laser with 42 mW, which can result in a better LOD for the former group. The intracavity power of Li et al. is 9 W at two foci and 80 W for Wang et al. Keeping in mind the 5 times higher integration time and 2 foci, the results for both groups are logical. The setups also differ in their collection geometry as Li et al. use $90^{\circ }$ detection with an additional collection mirror, and Wang et al. use a forward detection. In Table 6, LODs for hydrocarbons with different CERS setups are shown.

Here, low LODs in the range of just a few ppm have been accomplished. Like before, Li et al. have achieved the lowest LOD here for C${}_{2}$H${}_{2}$, as well as C${}_{2}$H${}_{4}$. Furthermore, in Table 7, more LODs for different analytes are shown.

The showcased LODs are all in the range above 100 ppm. Compared with the hydrocarbons, these are rather high and not compatible with specialized sensors. Here, Hippler et al. measured an LOD of 100 ppm for H${}_{2}$S at 1 bar total pressure but mentions that, in detection at 100 bar, which is typical for gas storage facilities and gas processing plants, the LOD would be further reduced to 10 ppm [102]. As state of the art commercial sensors are capable of sub-ppm levels of detection [6], CERS sensors need to improve their capabilities. Furthermore, this reinforces the previous statements, that one advantage for CERS is its multi-gas analysis capability, whereas, if the task is single gas measurements, other sensor types are better suited.

## 6. Comparison with State of the Art Sensors

In the previous section, CERS sensors and setups and their capabilities were presented and compared to each other. In this section, we compare CERS to other state-of-the-art gas sensor techniques in regard to their sensitivity, as well as their utility. Nonetheless, only a fraction of the existing sensor types can be presented, since the entirety would go beyond the scope of this review and would need a review by itself. As already mentioned in the introduction, gas sensors consist of a wide variety of techniques and methods. They can be roughly separated into the following categories: solid-state sensors, electrochemical sensors, and optical sensors, although a merging of different techniques in a sensor can also occur.

Optical sensors themselves can be divided into several subcategories. Besides Raman sensors and their various enhancement methods, optical sensors offer a diverse spectrum of methods. One such method is photoacoustic-based gas sensing. The effect on which this technique is based on, has already been used since the 19th century [18]. By modulating light intensity, a periodic pressure variation is generated, which can be detected by microphones. The sound which is measured is thereby generated by the analyte itself. The review of Palzer [21] showcases this method and its state of the art. Here, CO${}_{2}$ can readily be measured in the parts per million range, with limit of detections as low as around 8 ppm [103]. Compared with the showcased CO${}_{2}$ LOD of CERS of 130 ppm, photoacoustic sensors achieve a lower LOD. Yet, to be able to use photoacoustic sensors, several requirements need to be fulfilled. Dependent whether indirect or direct photoacoustic spectroscopy is used, the emission spectrum of the excitation light source needs to coincide with the absorption spectrum of the analyte to enable absorption and subsequently of the optoacoustic signal. With complex analyte matrices and broadband light sources, such as thermal emitters, this may also yield the problem of low selectivity. Here, photoacoustic spectroscopy faces bigger challenges compared to CERS. An already established gas spectroscopic analysis method is non-dispersive infrared absorption spectroscopy (NDIR). Here, absorption bands of gases in the infrared spectral region are used, so infrared radiation is absorbed. Utilizing the Beer-Lambert-Bogouer law, the concentration of the analyte can be calculated. This method, however, is highly dependent on the knowledge of the analyte, as the infrared source and optical filters have to be carefully chosen to accommodate the infrared absorption spectrum. Furthermore, due to being absorption spectroscopy it is limited by optical paths, where either small paths could lead to no significant change in intensity or longer paths could limit the dynamic range of the sensor itself. Dinh et al. have shown, in their review of state-of-the-art NDIR sensors, ways to handle optical paths by utilizing different shapes of measurement cells [5]. This has led to detection limits in the sub-ppm regime [104]. Nonetheless, non-dispersive infrared absorption spectroscopy has further limitations regarding possible gaseous analytes, as several, and most importantly, homonuclear gases, do not exhibit the necessary absorption bands in the infrared region.

Another large group of sensors are solid state sensors. Here, depending on the used materials, subcategories can be introduced. One example are semiconductor metal oxide gas sensors. Dey has given a more detailed review for these sensors [105]. The sensitivity of these sensors depends on a multitude of properties of the used nanoparticles and their manufacturing process. Generally, the LOD of these sensor types range in the ppm regime, whereas, for NO${}_{2}$, LODs in the ppb range were reported by Sahm et al. [106]. Recent advancements in this area are promising, as nanotubes and nanostructures offer miniaturization possibilities and more competitiveness. Yet, one large disadvantage of this class of sensors, for example, is its cross-sensitivity. Due to their working principle, several gases might affect the sensor, thus masking the real concentration of the analyte in study. To circumvent this, arrays of several sensors are being used.

One of the most commonly used industrial sensor type are electrochemical sensors. Guth et al. have highlighted in their review on the advancement of electrochemical sensors, the applications for these sensors [107]. Here, they have shown that, compared to laboratory setups, like spectrometer-based sensors, electrochemical sensors lack the precision and detection limits but are low maintenance and relatively cheaply available. Furthermore, recent advancements in nanotechnology pave the way for faster electrochemical sensors. To give a full overview for these sensors is not possible, as its field is too broad.

In conclusion, various sensors offer different advantages and disadvantages. CERS itself offers the advantages of optical sensors as it has high selectivity while maintaining limits of detection usable for applications other than high precision measurements. Furthermore, challenges to transfer the laboratory setup to onsite and low cost setups have already been addressed [97,108]. The main advantage of CERS is the ability to measure unknown gas compounds, whereas other optical sensors, like NDIR, need to have a fixed set of filters and sources reducing the possible measured gas compounds. Furthermore, other classes of sensors, such as solid-sate sensors, face the disadvantage of cross-correlation with analytes other than their specific target analyte; thus, they have to be judiciously chosen beforehand. This makes CERS an attractive method for IoT applications, which will be further discussed in the next section.

## 7. IoT and Raman Spectroscopy

Feng et al. have discussed, in their review, the possibilities of smart gas sensing [109]. Here, they showcased the growing gas sensor market value and their respective expected growth in different field of use, like defense, environment, and medical, for example. Furthermore, more functions are expected from these sensor, as artifical intelligence and machine learning in sensor arrays and network is supposed to lead to automated process lines in the likes of Industry 4.0. Cavity-enhanced Raman spectroscopy offers various possibilities to be used regarding IoT applications, as it is possible to measure in-situ processes, where ordinary non-enhanced Raman spectroscopy would lack the power. Furthermore, CERS has the advantage of measuring multiple gases at the same time, rendering it possible to evaluate complex processes. As a good example, various groups were able to utilize CERS or FERS for breath analysis. Hanf et al. were able using FERS to analyze 27 nl of exhaled human breath, differentiating even minor differences in composition, such as different isotopes, e.g., ^14^N^15^N from ^14^N_2_ and ^13^CO${}_{2}$ from ^14^CO${}_{2}$ [66]. Schluter et al. were able to monitor anesthesia by using CERS and a signal evaluation procedure based on spectral soft modeling technique [92]. Due to a multipass cavity, 250 ms measurement times were possible, allowing to resolve a single breathing event. Furthermore, the system has demonstrated its ability for anesthetic gases. Another application for IoT is the in-line monitoring of gases in industrial applications. Sandfort et al. used CERS as one part of the quality control for food chain management [100]. As food is highly perishable, it is important to control parameters, such as temperature, humidity, and gas emissions. In this context, the CERS setup can be used to detect, for example, ethene, which is an indicator for fruit ripening. Furthermore, CERS is not bound to gas analysis, as Yang et al. were able to detect O${}_{2}$, CH${}_{4}$, and CO${}_{2}$ after 1 h of degasification of water [110]. The potential for CERS in IOT applications has also been shown in various publications of the groups of Torsten Frosch and Jürgen Popp. It is possible to do isotopic spectroscopy and also quantifying gas fluxes [111]. In this regard, Metcalfe et al. were able to monitor bacterial mixed sugar metabolism of E.coli by monitoring different CO${}_{2}$ isotopes [112]. Furthermore, Sieburg et al. have showcased onsite CERS measurements [108]. Here, this group has investigated gas exchanges in 70 m depth, a timeframe of six months seasonal, and thereby monitored the groundwater levels and microbial activity.

## 8. Summary and Outlook

In this short review, we have shown the possibilities that Raman spectroscopy poses as an analytical tool for gas spectroscopy. Furthermore, we have introduced CERS as a technique to enhance the naturally weak Raman signal to become competitive with commonly used gas spectroscopy methods, such as FTIR and NDIR or electrochemical sensors. FERS as a sub-category of CERS has been briefly introduced, and some examples are discussed. Different setups of CERS are shown, and their respective advantages and disadvantages are showcased. Furthermore, commonly used locking-methods to keep a resonant cavity are briefly introduced. In the last part, the enhancement of the Raman signal for different publications are shown and compared regarding their used geometry and whether a multi-pass cell or a resonant cavity were used. Additionally, the benchmark for CERS setups in the form of the LOD regarding different CERS setups were compared. Here, CERS shows its potential to trace gases in the ppm regime. The possibilities of Raman spectroscopy, and especially CERS, for IOT were presented with some examples from the last years. In conclusion, CERS has the potential to be used in different applications. It can be further advanced by using microcavities, thereby utilizing quantum electrodynamical effects, such as the Purcell effect, to further enhance the Raman signal [113,114].

## Figures and Tables

**Figure 1 sensors-21-01698-f001:**
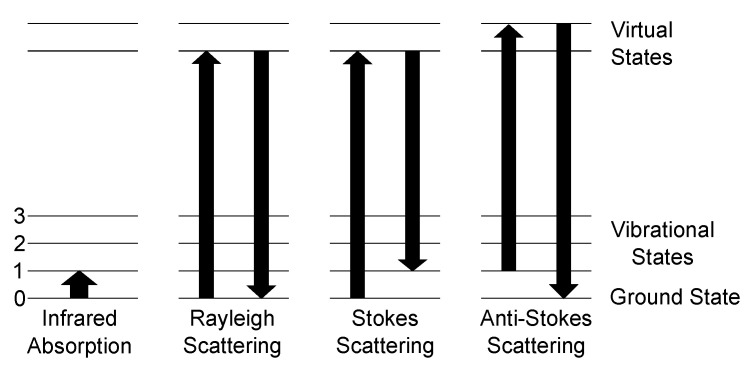
Schematic of the possible scattering processes between a photon and a molecule and ground state 0 and three vibrational states 1–3. Left: Infrared absorption process is governed by absorption of a photon with the energy of equal to the difference between the ground state and a vibrational state. Rayleigh scattering marks the elastic scattering process as energy is neither gained nor lost. The Raman scattering processes are denoted as Stokes and anti-Stokes scattering. Stokes scattering leads to a red-shift of the scattered spectrum with respect to the Rayleigh-line, since the final state is a higher vibrational level than the ground state. Anti-Stokes, on the other hand, starts at a higher state and ends at the ground state, resulting in a blue-shift of the spectrum.

**Figure 2 sensors-21-01698-f002:**
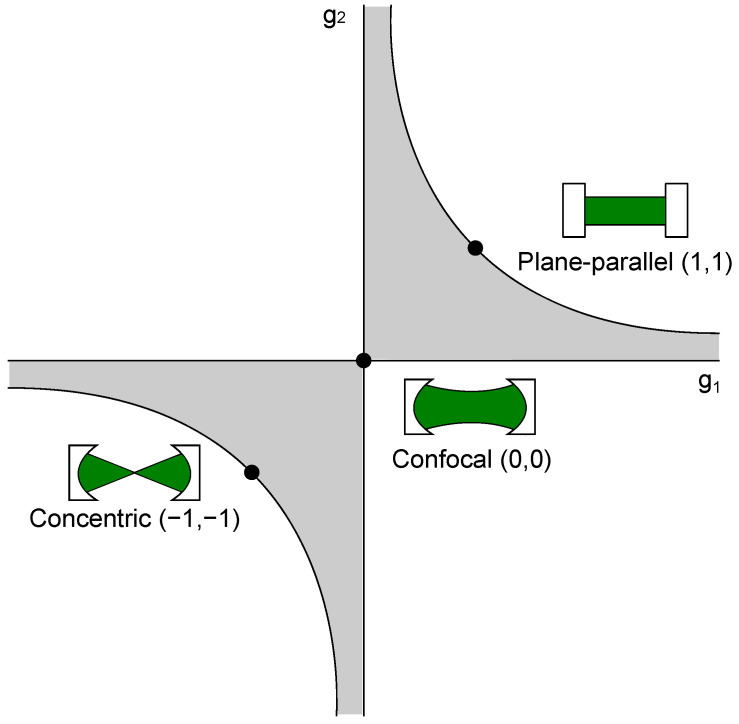
Stability condition of optical resonators. Exemplarily, the most commonly used resonator types, i.e., concentric, confocal, and plane-parallel, are highlighted in the diagram. The gray shaded area shows the regions where the stability criterion (Equation (Equation 13)) is fulfilled.

**Figure 3 sensors-21-01698-f003:**
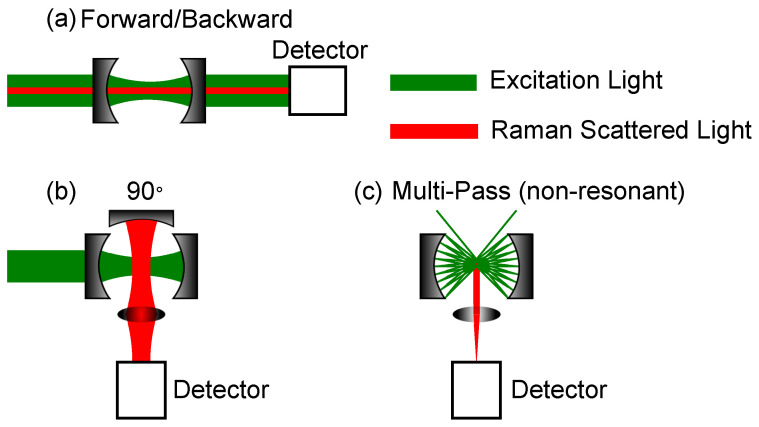
Most common geometries for cavity-enhanced Raman spectroscopy (CERS)-setups. The mirrors’ curvatures can be concave, as well as plane. (**a**) Measurement of Raman scattering in forward or backward direction. (**b**) Measurement of Raman scattering $90^{\circ }$ direction. (**c**) Measurement of Raman scattering using a non-resonant multi-pass cell. The detection is in a $90^{\circ }$ angle.

**Figure 4 sensors-21-01698-f004:**
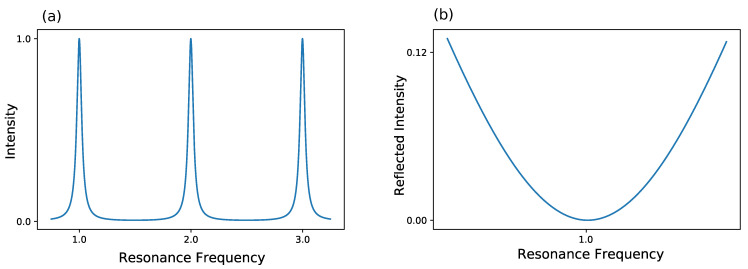
Behavior of a lossy Fabry-Perot resonator. (**a**) The intensity inside the resonator is shown. The intensity shows maxima at the cavity’s resonance frequencies and decreases away from these resonances due to destructive interference. (**b**) The reflected intensity is shown. At resonance frequency, the reflected intensity is dropping to zero.

**Figure 5 sensors-21-01698-f005:**
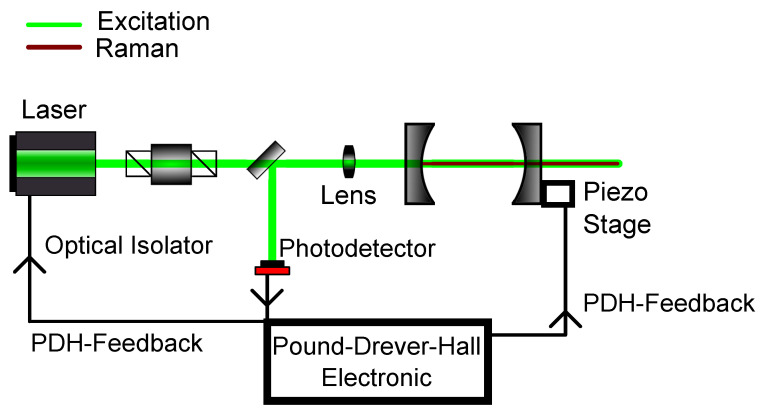
Schematic setup for Pound-Drever-Hall (PDH) modelocking. The reflected light of the first cavity mirror gives the necessary feedback to either change the cavity length with the use of the piezo stage or to modulate the laser frequency itself.

**Figure 6 sensors-21-01698-f006:**
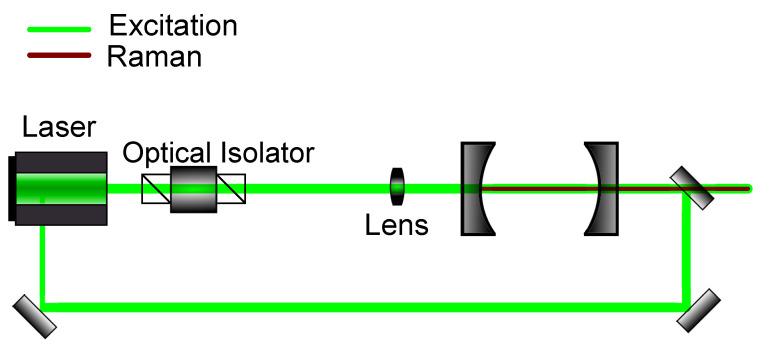
Schematic setup for optical feedback frequency-locking. The emitted light out of the cavity is injected back into the source laser to enforce a resonant frequency.

**Figure 7 sensors-21-01698-f007:**
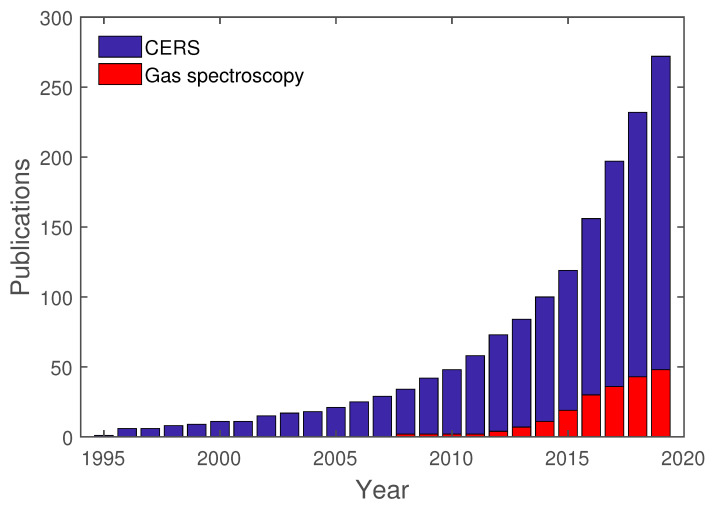
Publications regarding CERS, in general, and cavity-enhanced Raman gas spectroscopy, in particular.

**Table 1 sensors-21-01698-t001:** Comparison of the different enhancement methods mentioned in the text.

Enhancement Method	Methodology	Advantages	Disadvantages
Resonance Raman	Electronic transitions of specific analyte are stimulated	Enhancement of up to $10^{6}$, mostly used with chromophores	Wavelength-tunable laser (e.g., dye laser) needed
Coherent anti-Stokes Raman	Four-wave-mixing	No Fluorescence, also non-Raman active transitions possible	Two laser sources necessary, at least one tunable
Surface-enhanced Raman	Plasmonic enhancement	Large enhancement factors, detection down to single molecule level	Distinct nanostructured surface morphology needed, works only when analyte close to surface
Fiber-enhanced Raman	Light guiding and confinement	Long interaction lengths (thus enhancements) can be achieved while still compact	Only for gaseous and fluidic analytes

**Table 2 sensors-21-01698-t002:** Optical cavity locking methods and their advantages and disadvantages.

Locking Method	Advantages	Disadvantages
PDH	flexibility to either vary the laser frequency itself or the cavity length	need for high cost electronics
Optical feedback frequency locking	easy to implement via back reflection	needs specific semiconductor lasers

**Table 3 sensors-21-01698-t003:** Raman gains of non-resonant multi-pass cells.

Group	Geometry	Additional Collection Mirror	Raman Gain
Li et al. [90]	Near confocal	Yes	45
Utsav et al. [91]	Near confocal	No	20
Schlüter et al. [92]	Near confocal	Yes	18.52
Schlüter et al. [92]	Plane Mirror geometry	Yes	10
Petrov et al. [93]	Near concentric	No	20
Wen et al. [94]	Four mirror setup	Forward detection	9

**Table 4 sensors-21-01698-t004:** Raman gains of resonant cavities.

Group	Locking	Raman Detection	Raman Gain
Salter et al. [95]	Optical feedback loop	forward	833
Zaitsu et al. [96]	Not specified	forward	6000
Thorstensen et al. [97]	Optical feedback loop	foward	50
Friss et al. [98]	PDH	$90^{\circ }$	5900
Wang et al. [99]	Frequency-locking	forward	11.8
Sandfort et al. [100]	PDH	$90^{\circ }$	851
Wang et al. [80]	Frequency-locking	forward	2200

**Table 5 sensors-21-01698-t005:** Most common analytes and their limit of detection (LOD) of different CERS setups.

Group	CO${}_{2}$	H${}_{2}$	O${}_{2}$	N${}_{2}$
Li. et al. [101]	36 ppm	-	-	-
Li et al. [90]	16 ppm	-	12 ppm	-
Hippler [102]	-	140 ppm	-	1000 ppm
Wang et al. [99]	90.6 ppm	75 ppm	80.7 ppm	85 ppm
Sandfort et al. [100]	317 ppm	-	1412 ppm	3540 ppm
Wang et al. [80]	17.4 ppm	-	50.7 ppm	53.5 ppm
Wen et al. [94]	-	132 ppm	223 ppm	213 ppm

**Table 6 sensors-21-01698-t006:** Hydrocarbons and their LODs for different CERS setups.

Group	C${}_{2}$H${}_{2}$	CH${}_{4}$	C${}_{2}$H${}_{4}$	C${}_{2}$H${}_{6}$	C${}_{3}$H${}_{8}$
Li et al. [101]	12 ppm	6 ppm	-	-	-
Li et al. [90]	1.6 ppm	-	0.8 ppm	-	-
Salter et al. [95]	-	190 ppm	-	-	-
Hippler [102]	-	50 ppm	-	-	-
Wang et al. [99]	32 ppm	17.4 ppm	52.8 ppm	29.33 ppm	-
Sandfort et al. [100]	-	-	261 ppm	-	-
Wang et al. [80]	5.8 ppm	-	-	-	-

**Table 7 sensors-21-01698-t007:** Different analytes and their LOD for different CERS setups.

Analyte	Group	LOD
H${}_{2}$S	Hippler [102]	100 ppm
CO	Wang et al. [99]	130 ppm
H${}_{2}$O	Wen et al. [94]	109 ppm

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
