# Peer review of "A Short Review of Cavity-Enhanced Raman Spectroscopy for Gas Analysis"

_sensors, 2021, doi:10.3390/s21051698_

Round 1
Reviewer 1 Report
This is a largely well-written and timely review on enhanced Raman spectroscopy in relation to other gas anlytical tools. It collects different sensitivity claims from various groups and while a fair comparison would have to dig deeper into the details, at least it provides an entry point for the reader to do so. The advantages and shortcuts in comparison to competing techniques are also briefly summarized. The introductory part on the basics is mostly well-written but one weak spot jumped into my eyes. It refers to figures 3 and 4, where the ray geometries do not always make sense, in my opinion. In figure 3b, the principle of collection of Raman scattering in two orthogonal directions should either be illustrated with a plane mirror (which would then be a bit more consistent with the cartoonish detection arrangement), or else one should more realistically keep the concave mirror but then include a focal point of the scattered light and a collecting lens before the detector. More seriously, the ray geometry of the scattered light in figure 4 makes no sense at all. The light cone pointing downwards has a collection angle which does not match the collecting lens. Something went really wrong in this plot, I think. Probably most of the (semi)vertical red lines got somehow displaced. Clearly, the only advantage is the back-reflection, but that was already discussed in figure 3b. Perhaps most seriously, the statement that for gas Raman spectroscopy the spontaneous scattering is not directional is of course not correct. Even Rayleigh scattering is strongly directional for a given polarized laser excitation (as lasers typically are). It is most pronounced at 90° scattering if the polarization is perpendicular to the scattering plane and zero if it is parallel to the scattering plane. Similar considerations apply for Raman scattering, depending on the symmetry of the vibration. This whole section requires careful reformulation. I also don't understand how the total detection efficiency epsilon is an intrinsic property of the molecular species, if at the same time it is limited by the used detector.
There are some smaller stumbling blocks in formulations, where words are missing or other things complicate understanding. I just cite a few of them: "with each other as well with state-of-the-art sensors", "the ratio of number of molecules inside the illuminated volume per volume" (ratio and per together is duplicate), "Its based", "omitting therefore this study and have a closer look to the", "in this area promising".
After carefully removing these smaller and the above-mentioned optical flaws, I consider the review suitable for publication in Sensors.
Reviewer 2 Report
The presented work for review is interesting and contains a lot of information. While the chapters describing the principle of Raman spectroscopy and the methods of signal amplification are well described, the comparison of CERS with gas sensors is poorly written and requires improvement. Below are my remarks and comments:
line 19-21, correct, but in various cases there is no need to identify the components of the gas mixture. Such is the case with sensor arrays (electronic noses), which are increasingly used to distinguish gas mixtures in the environment, especially odors. It is also worth discussing this topic in the publication and using the literature on this topic:
1) Application of electrochemical sensors and sensor matrixes for measurement of odorous chemical compounds, Trends in Analytical Chemistry, 77, 1-13, 2016
2) On ‘Electronic Nose’ methodology, Sens. Actuators B Chem, 204 (2014), pp. 2-17
3) Applications and advances in electronic-nose technologies. Sensors 2009, 9, 5099–5148.
4) Monitoring techniques for odour abatement assessment. Water Res. 2010, 44, 5129–5149.
line 28-29, it is a limit of detection or a limit of quantification rather than a sensitivity. sensitivity is a static parameter related to the slope of the calibration curve. Please correct it.
line 47-56, it would be helpful to insert a table describing the pros and cons of each method here, it would make it easier for the reader to follow this work further.
line 68-114, chapter 2, I think that this chapter can be definitely shortened. The principle of Raman spectroscopy can be found in academic textbooks.
line 115-155, chapter 3, it is worth inserting a table describing the advantages and disadvantages of Raman reinforcement methods
line 215-248, chapter 4.2.2, here I also recommend inserting a table describing the advantages and disadvantages of each locking method.
line 325-327, it is difficult to comment that the CERS is a low level detection device, the values ​​of 100ppm for H2S are very high. Please comment on this, because commercial VOC sensors are capable of measuring below 1 ppm v / v. Please read the literature and comment on it, where is the technical advantage of CERS on gas sensors?
1) Currently Commercially Available Chemical Sensors Employed for Detection of Volatile Organic Compounds in Outdoor and Indoor Air, Environments 2017, 4(1), 21
2) Review of Portable and Low-Cost Sensors for the Ambient Air Monitoring of Benzene and Other Volatile Organic Compounds , Sensors 2017, 17(7), 1520.
line 328-386, chapter 6, this chapter is poorly described, many commercial sensors have LOD levels below 1 ppm v / v. So writing about CERS at the ppm level that they have very good parameters is out of date. I propose to modify this chapter thoroughly.
Reviewer 3 Report
The review titled: A short review of cavity-enhanced Raman spectroscopy for gas analysis is very well written and organized. The review describes general research, common setups, and achieved resolutions of cavity-enhanced Raman techniques.
The authors touch the very interesting branch with a high application potential that can solve drawbacks of the common gas sensors (electrochemical, semiconductor and NDIR – optical sensors) Simultaneously, the authors critically evaluate the limits of Raman spectroscopy for gas analysis (low numbers of molecules, low scattering cross section etc.).
Basic cavity concepts provides an overview of basic theory and experimental setup concepts; their advantages and disadvantages are showcased. Furthermore, commonly used locking-methods to keep a resonant cavity are briefly introduced.
In brief, I can only recommend the manuscript to be accepted for publication in Sensors. I have only two minor comments:
1) In the section 2. Raman theory, the authors explain basic physical background of Raman effect. The sentence on the line 70: Classically, the Raman effects can be described.... etc. One can get an impression that the description is based on classical electromagnetic theory, but authors present the description based on "basic quantum physics".
2) Missing references for equations 1 and 2.
Round 2
Reviewer 2 Report
The authors responded to all my remarks and comments. I agree with these answers. I also understood the authors' intention to describe Raman spectroscopy as a possibility to measure many gases. We can also see corrections made to the manuscript. Summing up, I recommend the manuscript for further stages of evaluation.